# Developing a Descriptive Sensory Characterization of Flour Tortilla Applying Flash Profile

**DOI:** 10.3390/foods10071473

**Published:** 2021-06-25

**Authors:** Sanjuana Rodríguez-Noriega, José J. Buenrostro-Figueroa, Oscar Noé Rebolloso-Padilla, José Corona-Flores, Neymar Camposeco-Montejo, Antonio Flores-Naveda, Xochitl Ruelas-Chacón

**Affiliations:** 1Department of Food Science and Technology, Autonomous Agrarian University Antonio Narro, Calzada Antonio Narro 1923, Saltillo 25315, Mexico; sjrdz96@gmail.com; 2Center for Research in Food and Development A. C., Unidad Delicias, Cd. Delicias, Chihuahua 33089, Mexico; josebuenrostro@ciad.mx; 3Department of Animal Production, Autonomous Agrarian University Antonio Narro, Calzada Antonio Narro 1923, Saltillo 25315, Mexico; uaaan_lacteos@yahoo.com.mx; 4Department of Planning, Autonomous Agrarian University Antonio Narro, Calzada Antonio Narro 1923, Saltillo 25315, Mexico; josedaniel.corona@gmail.com; 5Seed Technology Training and Development Center, Department of Plant Breeding, Autonomous Agrarian University Antonio Narro, Calzada Antonio Narro 1923, Saltillo 25315, Mexico; neym_33k@hotmail.com (N.C.-M.); naveda26@hotmail.com (A.F.-N.)

**Keywords:** flour tortillas, sensory characterization, flash profile, sensory analysis, generalized procrustes analysis

## Abstract

For any food, it is important to know consumption, preference, and the characteristics as quality parameters that are important to consumers of a product. The descriptive methodologies are an important tool to know the quality attributes of the products. Within these methodologies is the flash profile (FP), which is based on the generation of the distinctive attributes of the products without any expensive and time-consuming training sessions. The aim of this research was to study the consumption and preference of flour tortillas by consumers and to develop the descriptive characterization of the tortillas by using the flash profile method. The wheat flour tortillas used were two commercial and two handcrafted samples. Ten experienced panelists participated as the FP panel. The panelists generated 22 descriptors, six for texture, seven for appearance, five for odor, and four for flavor. These descriptors differentiate the samples of the flour tortillas. The panelists’ performance was assessed using the consensus index (Rc = 0.508). The first two dimensions of the Generalized Procrustes Analysis represent 83.78% of the data variability. Flash profile proved to be an easy and rapid technique that allowed the distinctive attributes of flour tortillas to be obtained.

## 1. Introduction

In the bakery industry, the flour tortilla segment has increased significantly in North America [1]. The tortilla is a thin Mexican flatbread made from corn or flour, used to accompany a great variety of dishes [2,3]. It is well known that after the Spanish domination began, wheat flour tortillas were developed in Mexico. The Spaniards and mestizos developed the wheat tortilla in order to satisfy their preferences. This type of tortilla possesses similar attributes and uses as the corn tortilla but with a different flavor [4]. Flour tortillas are basically prepared with flour, water, oil/butter, and salt. The mixture is kneaded until stretchable and cohesive doughs were obtained [5,6]. There are three methods to produce tortillas: hand-stretch, hot-press, and die-cut. The dough for the tortilla is obtained by using the ingredients mentioned above and certain types of flour. Each procedure includes a particular dough-forming procedure that requires unique flour characteristics [7,8].

Therefore, flour tortillas have various end-uses and characteristics. The hot-press procedure is today the most used (90%) in the total production of tortillas. The dough is cut into small balls of the same weight and shape, then they are crushed into discs through heated platens to obtain the tortillas [6]. Flour tortillas must have a soft and flexible texture during storage and are normally sold in retailed markets accessible to consumers of this product [8,9].

Stronger tortilla doughs that absorb a greater volume of water are obtained by the die-cut procedure. This procedure consists of passing the dough into the rollers so sheets can be formed and which are cut into uniform circles. The die-cut process is appropriate for a bigger production capacity. It is more efficient and generates lower-cost products compared with the hot-press process. Nonetheless, the tortillas made with this procedure are not as adhesive, soft as the ones made with the hot-press process, and the flexibility is lost more quickly during storage. Despite these disadvantages, these types of tortillas are used to prepare processed foods, such as enchiladas, burritos, and chimichangas [6,8,9].

The third procedure used to obtain flour tortillas is the hand-stretch method. The tortillas are generally thinner and firmer than the ones made with hot-press or die-cut processes. Unfortunately, for the production of tortillas by this method, more personnel are required, and the production speed is slower, which is why tortilla manufacturers do not use this method on a large scale [2,5]. For all the aforementioned of the three processes, the one that is most used for the production of flour tortillas is the hot-press, and for this reason, it is a subject of interest for investigation and processing of the tortilla [6,8]. 

For any of the flour tortilla-making processes, other ingredients, such as antimicrobial and antioxidant agents, among others, can be added to extend the shelf life and to improve the sensory characteristics demanded by consumers [1,2,10]. 

Yeverino et al. [11] determined the physicochemical characteristics of flour tortillas in the metropolitan area of Monterrey, Mexico. The authors found that ready-to-eat flour tortillas come from different processes, and there is a great variety in their composition. For this reason, it is important to give an additional value to the product for the consumer to choose to purchase it within such a competitive market. It is worth mentioning that, within the parameters measured by these authors, each and every one of them is perceived by the five senses. Color, for its part, is considered relevant for the case of sensory evaluation in the food industry because this property can cause a food to be accepted or rejected immediately by the consumer without even having tried it. In the case of appearance, it represents all the visible attributes of a food, and it constitutes a fundamental element in the selection of a food and is a deciding factor when purchasing products [12]. Therefore, the sensory characterization of food is important in the food industry. By using sensory descriptive methods, specific product attributes can be obtained and are consider quality parameters.

A common practice for some time in the food area has been to describe the sensory characteristics of products, either to identify the characteristics of the product, identify the result of the change of an ingredient or phase of the process, evaluate the shelf life, match the standard product, development of a new product or as a means of quality control [8,13,14]. An important and widely used tool in sensory evaluation is the sensory characterization of food. It allows obtaining and establishing qualitative and quantitative aspects of the perception by humans, and it makes it possible to measure the sensory reaction that the stimuli generated by a product [15]. 

Conventional descriptive methodologies allow obtaining sensory profiles, however, with certain limitations that delay investigations. Some of the limitations are related to the difficulty of measuring perception, training time, and the resources necessary to form and maintain the descriptive panel [13,16,17]. Flash profile (FP) is a rapid descriptive method related to the conventional profile, which generates and quantitatively evaluates the sensory attributes of products. The aim of FP is to provide a rapid and reliable sensory profile of a set of products according to their major sensory differences. This method relies on individual vocabularies and comparative assessments to shorten the time needed to perform the analysis [18]. The development of multivariate techniques for statistical analysis of data and Generalized Procrustes Analysis (GPA) has made this method very reliable and used in sensory laboratories [16,17]. GPA is a multivariate exploratory technique that involves transformations of individual data matrices to provide optimal comparability. The average of the individual matrices is called the consensus matrix. GPA uses individual scores to account for any variance. Since all the panelists evaluate the same samples, the samples remain constant and do not vary [18]. 

As a way to characterize and quantify sensory differences and similarities among products, conventional descriptive profile methodologies are generally applied. This requires the training of panelists and the rating of the products’ attributes based on a consensual vocabulary [6,16,17]. These methods provide a complete sensory profile, but unfortunately, it is expensive and time-consuming because of the long-term training of panelists in order to gain consistency and reproducibility of the attributes of the products. Flash profile analysis (FP) is an alternative, simpler, and less time-consuming technique that can be applied to investigate differences and similarities among samples [6,17,18,19] and based on Free-Choice Profiling (FCP) [16,18,19,20]. FP considers ranking by direct comparison of a set of simultaneous samples. The FP combines the individual attribute elicitation of the FCP method with a comparative ranking of samples on these attributes. In contrast to FCP, the assessors have to rate the intensity on each attribute of the products for each selected term, and that can be a different number of terms per assessor [21,22,23]. The comparison of all products at the same time is claimed to remove the need for product familiarization and training with the attributes, with the possibility to go back to previous assessments. The process of elicitation and training is thereby drastically reduced. The FP relies on experienced or expert panelists, which already have a considerable frame of reference for naming sensory perceptions. The FP method has been compared with conventional descriptive sensory profiling on products that are typical for an industry situation of brand comparisons [23,24]. 

Originally, FP was developed to be carried out by experienced panelists. The panel could be sensory experts and/or professionals who have developed sensory experiences with the products. For the number of panelists, the original FP method used eight panel members [24], but later applications of the method reported 6–12 panel members to be sufficient [24,25], and the maximum number of samples recommended to be given per session should be no more than 10 to 12, although some serving temperature and carry-over effects must be considered, whereas Delarue [17] stated that a panel of 4–5 members is the minimum to yield a stable configuration on sensory product differences. There is no need for training panelists to obtain by consensus the attributes and standard anchors for the evaluations [24,25,26].

Recent studies have also explored the application of the FP method with consumer panelists. In these studies, 24–50 consumers were considered sufficient [13,21,27,28]. Depending on the objectives of the study, the number of consumers could even range to 200. Consumer panels, however, require special explanations on the use of the method and instruction on using product attributes. The interpretation of the attribute space from consumer panels may also be difficult due to the extensive and unclear use of sensory terminology [26,28].

The use of the Flash profile technique is relatively new but has been applied to the analysis of a great variety of products, including dairy, meat, poultry products, snacks, bakery products, jams, juices, marmalades, among others food samples [24,25,26]. There are no published studies applying sensory descriptive analysis to flour tortillas.

The aim of the present study was to investigate the consumption and preference of flour tortillas by consumers and to generate a descriptive characterization for wheat flour tortillas by applying the Flash profile method.

## 2. Materials and Methods

### 2.1. Materials

Four wheat flour tortilla samples (two local retailers and two commercial brands) (Table 1), Reyna brand 12-ounce cups and Reyna brand, cornstarch-based rectangular trays (8.46 in length, 6.41 in wide and 0.90 in height), evaluation formats, pens, and napkins were used.

### 2.2. Consumption and Preferences of Flour Tortilla Consumers

A four-question survey was applied to 100 consumers in Saltillo, Mexico. The questions included age range, the brand of flour tortilla consumed, kind of flour tortilla preferred, and frequency of consumption. Panelists were included on the basis of 50 women and 50 men, age range between 20 and 45 years, being a flour tortilla consumer, and their willingness to participate in the survey. The answers were counted and expressed as a percentage.

### 2.3. Flash Profile

#### 2.3.1. Samples

Wheat flour tortilla samples were purchased from two local retailers and two commercial brands in Saltillo, Mexico. The two local manufactures of flour tortillas were labeled as “Bomberos” and “Super García”, and the two commercial brands were labeled as “Soriana” and “Tía Rosa”. The total number of samples was four. The diameter and thickness of each brand were as follow: “Bomberos” 5.91 in diameter and 0.08 thick, “Super García” 5.51 in diameter and 0.08 thick, “Soriana” 6.7 in diameter and 0.10 thick, and “Tía Rosa”. The tortillas were given at a temperature ready to eat (45 ± 5 °C).

#### 2.3.2. Panel

The panel was made up of ten trained panelists (seven women and three men, 22–25 years old) who are habitual consumers of flour tortillas. The panelists were recruited from staff and students of the Agrarian Autonomous Antonio Narro University. Their selection was based on experience in evaluating food products with descriptive sensory analysis and consumption of flour tortillas. All of them had previously participated in discriminative, consumer, and sensory profiling analysis. They were members of the trained panel of the Antonio Narro Autonomous Agrarian University and students of the engineering career in food science and technology. All were familiar with wheat flour tortilla attributes. However, they were not specifically trained in the evaluation of flour tortillas, and they had not evaluated flour tortillas previously. Each panelist generated from 12 to 22 attributes adding a total of 187 sensory descriptors. The attributes of the samples of flour tortillas generated corresponded to appearance, texture, smell, and taste.

A survey with 4 questions was applied to 100 consumers, of which 50 were women and 50 were men, to determine the consumption and preference of the flour tortilla. The age range of the surveyed population was 20 to 45 years, 65% of the population was within the age range of 20–25 years, 19% people over 45 years, 9% people between 25 and 35 years old, and 7% people between 25 and 35 years old.

#### 2.3.3. Sensory Evaluation Procedure

By following Montanuci et al. (2015) [29], the sensory evaluation analysis was performed. The sensory evaluation of the flour tortillas was performed in the Sensory Evaluation Laboratory of the Food Science and Technology Department in the Autonomous Agrarian Antonio Narro University (UAAAN) in Saltillo, Mexico. Their facilities include individual booths with comfortable cushioned chairs, lighting, and room temperature control. Each flour tortilla was heated on a non-stick pan on the stove; first side of the tortilla was heated for 6 s and then turn facedown, heated for another 6 s, then it was placed in the plastic bag with a three-digit random code and placed in the tortilla container. Judges were instructed to rinse mouths with water at room temperature before and after samples. All samples were presented in each session of the Flash profile methodology. Before the test, panelists received a brief explanation and outline of the Flash profile method with a description of the fundamental phases. In the first session, the panelists were instructed on the generation of attributes evaluation of attributes of appearance, odor, flavor, texture, and also any sensation in the throat after swallowing (residual). All samples were given simultaneously to the panelists in a randomized order. The number of attributes that the panelists could generate was unlimited, and they were not to use hedonic terms (e.g., dislike, like, etc.) and were to focus on attributes that characterized the global set of flour tortillas, not individually. In the second session, it was required that the panelists read the attribute list, revise and refresh their final list. The panelists continued to place the flour tortilla samples in rank order on the ranking scale, anchored at the left with a minimum of the intensity of the corresponding attribute perceived and at the right with the maximum intensity of the attribute perceived. The time length for each session was different for each panelist (30–60 min).

#### 2.3.4. Data Analysis Method

The results were analyzed through Generalized Procrustes Analysis (GPA) using XLSTAT (Addinsoft, New York, NY, USA) in order to minimized differences between panelists of the sensory analysis. From the data matrices of the profiling assay, GPA calculates a consensus, and the generated plots demonstrate the differences and similarities of the flour tortilla according to the graphic interpretation.

## 3. Results and Discussion

### 3.1. Consumption and Preferences of Flour Tortillas by Consumers

Consumers were asked what brands of tortillas they consumed were. The following were considered within the proposed variants: flour tortillas without brand, tortillas made with La Perla flour, and commercial tortillas “Tortillinas”. According to the surveyed population, 52% consumed wheat tortillas without a registered brand, while 23% preferred to make their tortillas, 20% purchased the commercial brand “Tortillinas”, and 5% consumed the brand name La Perla.

According to the kind of tortilla that was consumed, 74% of the surveyed population consumed white flour tortillas, 17% whole wheat flour, and 9% both.

Regarding the frequency of consumption, the results of the surveyed population, 62% consumed it three times a week, 14% consumed it once a day, 9% every day, 9% occasionally, and 6% consumed flour tortilla more than once a day.

The consumption of the flour tortilla is a deeply-rooted phenomenon in the north of the country since it is where most of the wheat is harvested. Tortillas are an extraordinary alternative to bread, which can be consumed alone by making a roll or cut into 4 or 6 pieces to collect food, or they can be used to wrap different stews [13]. Like all developed food, it is important to determine the chemical and sensory composition. For this, it is important to establish a standardized production process, as this influences the final characteristics of the product.

### 3.2. Flash Profile Methodology

The residual variance for each manufacture after performing GPA is shown in Table 2. The manufactures which had the lowest residual had the most consensus for the attributes provided by the panelists. These manufacturers were Soriana and Super García. The manufacture Bomberos had the highest residual, therefore the least attribute consensus followed by Tía Rosa.

In relation to the residuals per panelist, the results demonstrated that panelists 2, 3, 4, 5, 6, and 9 conferred an exceptional agreement about the attributes that have been used (Table 3). The residuals of each panelist after the Generalized Procrustes Analysis (GPA), showed that residuals were higher for panelists 4 (48.223%), 6 (26.844%), and 9 (76.507%), which indicated that the panelists were further from the consensus when compared with the other panelists. The scaling factors for each panelist are given in the third column of Table 3. Consequently, the panelists whose scaling factors were higher than 1, panelist 1 (1.990), 7 (1.993), 8 (1.452), 9 (1.123), and 10 (1.750) showed that their configurations had been stretched to reach the consensus, due to the fact that they rated some attributes with the lower part of the scale, while the panelists whose scale factors were lower than 1, panelist 2 (0.752), 3 (0.724), and 6 (0.696), scaled-down their configurations to reach consensus, as a result of the used the narrow part of the ranking scale. Panelists 4 (1.018) and 5 (0.944) obtained scaling factors close to 1, so their configurations have not undergone further modification in the scaling stage of the GPA. According to Ser [29], some bias may occur as the panelists evaluate the scale. However, the GPA resolves this situation.

The consensus index (Rc) was used to determine the consensus of the panelists. The evaluation of the consensus of the panelists was carried out using the permutation test with a *p* < 0.0001, taking as the consensus index the percentage of the consensus variance in the total variance (Rc), while in the determination of discrimination, the correlation coefficient was applied to reveal which sensory terms were correlated with the principal components (CP) since the sensory term is important for the characterization of the samples in the sensory space [10,13,30]. The Rc for the panelists was 0.508 (50.8%). This value indicated a positive correlation; that is, an adequate consensus for the flour tortilla samples was generated in the performance of the panelists. This value was close to the values reported by Hernández et al. [31] in the evaluation of cheeses with untrained judges (Rc = 55.3%) and Silva et al. [32] in the evaluation with consumers of a mushroom-based sauce (Rc 56.0%); however, it was lower than that reported by Ramírez et al. [33] in the sensory characterization of chip-type fries (Rc = 78.1%) and Wu et al. [34] in the sensory description of yogurt using the Free Choice Profile (Rc = 77.7%).

Figure 1 shows the location of the four products and the consensus, evaluated by the ten panelists, each one in a different quadrant corresponding to F1 and F2 dimensions (83.78%). It is clearly represented that the panelists managed to group the samples of flour tortillas in Bomberos and Soriana in opposition to the samples Super García and Tía Rosa. For this reason, the samples Bomberos and Soriana were perceived and described with a larger number of attributes for odor, taste, appearance, and mainly texture, while Super García and Tía Rosa had fewer attributes regarding appearance, texture, taste, and odor (Figure 1. The possible differences found could be due to the characteristics of the products by itself and that panelists perceived a greater number of attributes from the manufacture of flour tortillas Bomberos and Soriana compared to Super García and much less from Tía Rosa.

Ser [29] applied the GPA with descriptive data of lamb meat, and 26 panelists were able to group the effect of preslaughter diet/management system and fasting period on sensory attributes, with a 66.74% variability of the data. Puma and Nuñez [13] used the FP method to sensory characterize four samples of hot-dog with nine trained panelists. The data showed that the panel was able to group the four hot-dog samples with a 92.85% variability.

The result of the Flash profile analysis represented in two dimensions (59.21% for the first dimension and 24.57% for the second dimension) explained 83.78% of the variability of the data (Figure 2). The consensual configuration of the flour tortillas samples and the attributes obtained by panelists after GPA are presented in Figure 2. Among the 22 attributes that present the highest frequency of citation, rollability, thickness appearance, softness, flour odor, sweetness, roughness, brown spots appearance, flour flavor, elasticity, salty odor, homogeneity, rubberiness, buttery odor, and thin appearance were the attributes that best correlated with the first dimension. The attributes that best correlated with the second dimension were sponginess, sweet odor, greasy flavor, greasy appearance, raw appearance, and toasted appearance.

The panelists characterized Bomberos flour tortilla as flour flavor, roughness, salty odor, brown spots appearance, toasted appearance, and raw appearance. Soriana’s flour tortilla was described as rubberiness, thickness appearance, flour odor, elasticity, greasy odor, greasy flavor, and sweet odor. The Super García was characterized as thin appearance, buttery odor, rollability, greasy appearance, and sponginess. The Tía Rosa flour tortilla was described by homogeneity, softness, sweetness, and salty taste. According to Kobayashi and Benassi [35], only the descriptors with correlation coefficients equal to or greater than 0.60 and at the same time cited by several panelists should be used to visualize the relationships between the samples and the attributes. The differences that the panelists perceived for each flour tortilla could have a relation with the ingredients used in their elaboration and handcrafted, semi-mechanized, or industrialized processed.

The variability percentage of this study (83.78%) was higher than that obtained by Rason et al. [36] in the sensory characterization of traditional dry sausages (78.0%), Ramírez et al. [37] in the sensory characterization of fish burgers (83.23%), Ramírez et al. [38] in the sensory characterization of smoked shrimp (82.39%), Gamboa et al. [39] in the determination of sensory attributes of Manchego type cheese during ripening (72.69%), Dairou and Sieffermann [24] in the comparison of 14 jams characterized by QDA^®^ and Flash Profile (69.0%) and Silva et al. [32] in the characterization of mushroom-based sauce (82.60%).

## 4. Conclusions

The majority of the analyzed population preferred white, homemade, and unbranded wheat flour tortillas and the frequency of consumption was three times a week. The Flash profile methodology is a useful tool to characterize flour tortillas. It allows quick access to product positioning and its distinctive attributes. The methodology provides information that differentiates the samples, and as well, it does not require the training stages of a trained panel for conventional descriptive methods, reducing the time and cost of descriptive tests. The panel consensually generated the 22 descriptors with the Flash Profile method for flour tortilla samples. The descriptors obtained are an initial step to detect and help to understand the important sensory attributes for the consumer of flour tortillas, which will allow their measurement and definition by a trained panel with the use of the quantitative descriptive analysis (QDA), providing a complete, detailed and accurate information about this product.

## Figures and Tables

**Figure 1 foods-10-01473-f001:**
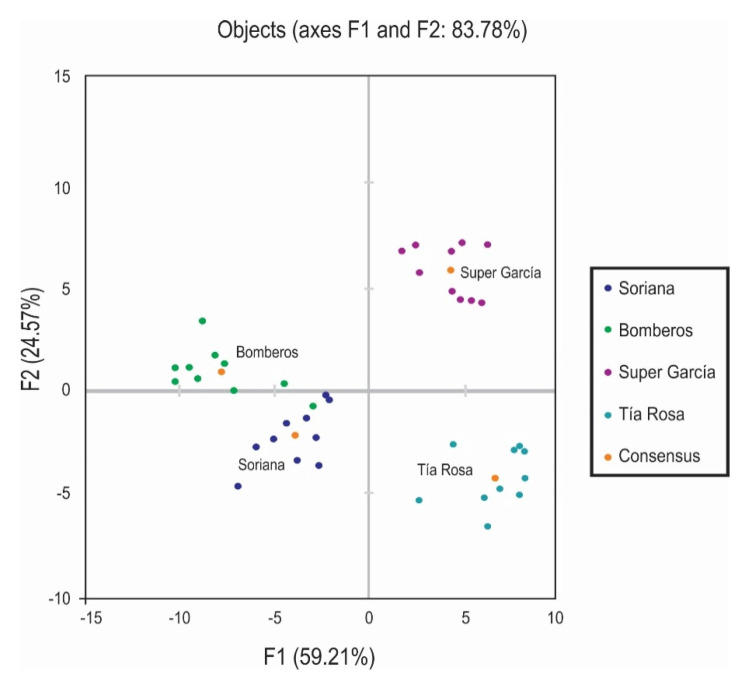
Sensorial space and consensus of the four manufacturers of flour tortillas with an 83.78% of variability of the data from GPA.

**Figure 2 foods-10-01473-f002:**
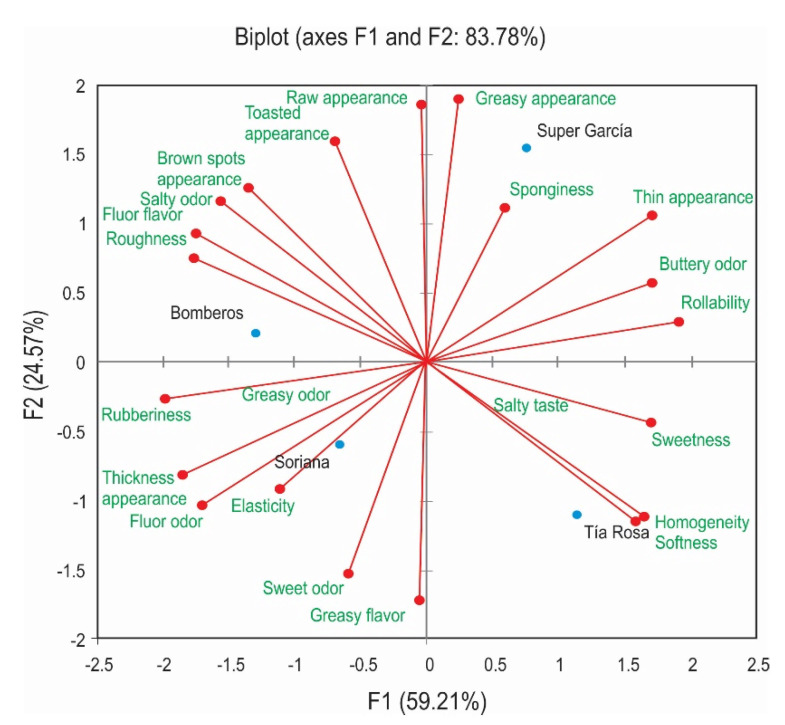
Biplot of the four flour tortillas, manufacturer Bomberos, Super García, Soriana, and Tia Rosa, and the lexicon used to describe them at the first and second dimensions of the General Procrustes Analysis (GPA) from Flash profile test.

**Table 1 foods-10-01473-t001:** Ingredients used and declared by the brands of flour tortilla studied.

Flour Tortilla Brand	Ingredients
Bomberos	Wheat flour, iodized salt, baking powder and shortening.
Super García	Wheat flour, iodized salt, baking powder and shortening.
Soriana	Wheat flour, iodized salt, baking powder, shortening and milk.
Tía Rosa	Wheat flour, partially hydrogenated vegetable fat, iodized salt, sodium aluminum sulfate, monocalcium phosphate, mono and diglycerides, vegetable oil, enzymes, sugar, calcium propionate, sorbic acid, active soy enzymes, calcium peroxide.

**Table 2 foods-10-01473-t002:** Residual variance for each wheat flour manufactures from GPA of Flash profile.

Wheat Flour Tortilla Manufacture	Residual (%)
Soriana	54.096
Bomberos	71.605
Super García	54.325
Tía Rosa	63.609

**Table 3 foods-10-01473-t003:** Residual variance, scaling factors, and the variation percentage explained by the first two principal components of Generalized Procrustes analysis (GPA) for each panelist.

Panelists	Residuals, %	Scaling Factors	F1, %	F2, %
1	20.564	1.990	58.845	30.900
2	7.663	0.752	53.765	30.191
3	12.170	0.724	73.544	16.103
4	48.223	1.018	55.879	34.258
5	4.965	0.944	52.535	29.109
6	26.844	0.696	39.992	35.804
7	18.550	1.993	81.614	11.079
8	14.024	1.452	78.233	12.713
9	76.507	1.123	12.062	38.078
10	14.125	1.750	66.680	13.429

F1. First principal component of GPA; F2. Second principal component of GPA.

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
