# Peer review of "Developing a Descriptive Sensory Characterization of Flour Tortilla Applying Flash Profile"

_foods, 2021, doi:10.3390/foods10071473_

Round 1
Reviewer 1 Report
The work presents the development of a sensory vocabulary for wheat flower tortillas, using the Flash profile method. It shows how a new sensory vocabulary can be created. It is a thorough study, although there appear some loose ends that need repairing.
I understand that the main objective appears to be the generation of a vocabulary, which could be made somewhat more clear. If I’m mistaken, this may also call for somewhat more clarity concerning the main objective of the study.
The section where the authors introduce FCP, needs rewriting. I feel their presentation of FCP is not correct, and incomplete.
Four products may be a meagre base to perform a GPA on. I feel the authors may need to address this issue.
Several of the figures appear superfluous, they do not add information and can very well (perhaps even better) be replace by tables, of by just mentioning the few numbers in the text: Figures 1 and 3 are not needed, perhaps the same is true for figures 2 and 4.
The ‘consensus index’ needs better introduction. It is now not clear to me what index the authors mean. As a result Figure 7 is also not clear.
The type of randomisation/permutation test needs clearer introduction, if this is indeed what was intended (compare ref [29]). A resulting p-value of the ‘consensus index’ is not provided (other than in the abstract?). This part of the methodology remains rather unclear, and needs rewriting.
Is it possible to attach a meaning to the two dimensions (see Figure 8)? E.g. based on the loadings (correlations of attributes with the dimensions)?
Title
2: ‘Consumption, preference’ are mentioned in the title, but hardly figure in the paper. I’d suggest taking them out of the title, and changing the title into e.g. “Developing a descriptive sensory characterization of flour tortilla applying Flash profile”, as that is the main matter presented in the paper.
Abstract
24: remove ‘d’ from ‘characterized’
25: insert ‘method’ after ‘the flash profile’
29: listing a p value in an abstract may be too much detail for an abstract
Main text
43-44: sentence scrambled? Please check/rephrase
51: ‘die-cut procedure’: will readers know what is meant (I don’t), perhaps it’s jargon?
67-68: remove ‘chemical and natural’, it does not any give any information; the source of the agents that can be added is outside the scope of the paper
76: ‘of the human beings’, should be replaced by ‘by human beings’
77: part of this sentence scrambled. One could omit ‘beings, and it makes it possible to measure the sensory 76 reaction that the stimuli generated by a product’ from this sentence
85: GPA is an established statistical method, but ref 14 and 15 may not be the appropriate (statistical) introduction texts to this method.
86: is ‘As’ missing as first word in this sentence?
94: ref 19 does not sound as it introduces the FCP method
94-86: the description given does not describe the FCP (Free Choice Profiling) method. The authors should rewrite this, and provide the proper references to FCP methodology. The authors write that FCP ‘does not consider rating, but FCP may include rating (freely, individually, selected profiling).
95-96: sentence scrambled?
106: remove ‘consumers’
112-113: what (metric) unit are these numbers in?
148: the authors write ‘on the evaluation of attributes’, but they may rather mean ‘on the generation of attributes’? Evaluation (scoring the intensity of) the attributes is done later in the process.
153: ‘required’ instead of ‘enquire’?
155-156: it reads: ‘a minimum of the corresponding attribute’; perhaps ‘a minimum of the intensity of the corresponding attribute’, is more clear?
156: similarly: ‘maximum intensity of the attribute perceived’, instead of ‘maximum attribute perceived’.
156-157: to have ‘ties’ (that occur in the scoring), and ‘rest brakes’ (that occurs with the panelists) like this in one sentence reads a bit strange.
160: GPA is introduced here without proper reference to the method. I suggest the authors refer to either a statistical paper introducing it, or to a paper that introduced the method in sensory analysis. Two of these are:
Dijkjsterhuis, G.B., Gower, J.C. (1991/2). The Interpretation of Generalized Procrustes analysis and Allied Methods. Food Quality and Preference, 3, 67-87.
Dijksterhuis, G.B. (1996). Procrustes Analysis in sensory research. In: T. Næs, E. Risvik (eds.), Multivariate Analysis of data in sensory science. Amsterdam: Elsevier Science Publishers.
but there probably are much more recent papers introducing GPA to sensory analysis data.
201-203: scrambled sentence?
208: ‘that’ and ‘received is’ can be deleted from this (rather strange) sentence.
209-213: scrambled sentence.
221-222: 12 to 22 attributes are mentioned, with 10 panelists. The total number seems to be 220, but then all panelists must have generated 22 attributes. How is this possible?
223: ‘performed by’ should be ‘performing’
224: What do the authors mean by ‘criteria’; this does not become clear, nor how the authors can draw this conclusion. Do we know the criteria by which panelists assess the tortillas? We know the attributes that were elicited, but these are not the ‘criteria’, as the attributes are not ‘similar’ (at least not in name), because the FCP method was employed. Or to the authors conclude that the different attributes point at similar ‘underlying’ criteria? In that case the loadings should inspected first, in order to allow this conclusion.
The low residuals (per panelist, not per product) may indeed indicate similarity of the individual panelist configurations. But how can we conclude residuals are low? See remark on figure 5.
Figure 5: what is on the y-axis (percentages? How are the resicuals scaled?
233: remove ‘evaluation of the ‘
234: replace ‘where the’ by ‘showed’
234: ‘residuals’ instead of ‘residual’
Figure 6: same as previous figure: how are the residuals scaled, it’s hard to judge their meaning along the y-axis.
241-249: ‘consensus index’: it’s not entirely clear what is meant by this index. It needs explanation. It looks like it may be a percentage of Variance Accounted For, but I’m not sure. A clearer reference to [29] is needed, as they also use Rc in their paper.
As the Rc-value does not become clear, Figure 7 is also not clear. Plus as it contains almost no information, it may not be needed as a figure, but it may be suffice to mention the numbers in the text (provided Rc-value is clarified, and what’s in Figure 7 is clarified). If the idea is that this figure shows a permutation or randomisation test based p-value (like figure 1 in [29]) this must be more clear, and this type of test must be introduced more clearly. The Figure 7 appears mistaken, as there are only two bars visible, alternatively the randomisation procedure may not have worked properly?
262: In figure 5, the residual for Soriana is also low. This text does not seem to tally with Figure 5?
276: ‘obtained´ instead of ‘obtain’
278: ‘thickness’ intead of ‘thinkness’
Author Response
I´m attaching responses, thank you

Reviewer 2 Report
This manuscript entitled "consumption, preference and descriptive sensory characterization of flour tortilla applying Flash profile " is demonstrated the tortilla sensory property using a flash profile with Generalized Procrustes Analysis.
The whole story is well demonstrated. However, the current interpretation of preference data is not deep. Thus, I request the additional interpretation of the results.
Results and discussion
In Fig.1 -4, the consumer's consumption ad preferences were well demonstrated. However, this data is not well interpreted by 10 panelist's sensory tests. Generalized Procrustes Analysis is performed only in the 10 panelist's analyses. Thus, the author should make discussion on a larger scale test using the results of the consumer preference test.
Author Response
Thank you for your comments, I´m attaching the changes made according to suggestions of all reviewers.

Reviewer 3 Report
General comments:
The subject of the study is interesting.
I appreciate the Authors approach of consumers preferences and evaluation a quality of wheat flour tortillas using a Flash profile analysis.
However the scope of the work is interesting, there are some points in the manuscript that need deep improvement. Some parts of the manuscript should be modified, completed and commented or simply removed. The study has great potential, but needs major improvement.
Specific comments:
Ad. Introduction
In lines 95-96 Authors mentioned that there is no need of training panelists to obtain by consensus of the attributes for the FP analysis and then in the section 2.3.2 they put an information regarding using only a trained panel for evaluations. Why?
The Introduction does not convinced me why this method was used in this study. Please extend this issue
Ad. Materials and Methods
Section 2.1.
Please complete the missing information:
The full recipe composition should be provided in this section.
Section 2.2
The characteristic of the consumer group should be given.
2.3.2 Samples
There is a lack of important information. How the samples were prepared, what about size for individual evaluation? Were they warm or cold? What about temperature?
Section 2.3.3
See lines 140-143 it is about samples.
Lines 144- 152 – it concerns “procedure”
The reader does not know what a scale it was.
Ad. Results and discussion
Lines 166- 170 and Fig. 1 - it is methodological aspects. It is a group characteristic which was involved in this study. There are no results!
The results begins with the line 173.
Fig. 2 and Fig 3 – should be removed. It is enough to describe the results.
Lines 197- 207 this part is good for introduction.
Lines 215- 222- it should be a part of methodology.
Please remove the Fig. 7.
Fig 8 is of very low quality; almost unreadable. Please correct it.
Lines 281-282 and 284- 285 - such a presentation of the results is unacceptable
Please improve this carefully.
The Fig 9 is very interesting but not fully described and discussed.
Please note that each figure should clearly communicate to the reader what the values mean. There is no descriptive legend for these.
Author Response
I´m attaching the responses to your suggestions, thank you

Round 2
Reviewer 3 Report
The Authors corrected their work in line with the reviewer's comments.